# Pan-cancer analysis reveals *TREM1*+ PMN-MDSCs as critical regulators of immune suppression and tumor microenvironment remodeling

Yangjie Cai[1,7], Shanhang Li[2,7], Hening Li[3,7], Zhuan Zou[4,5,7], Xinda Zheng[1], Haijun Tang[4], Mingxiu Yang[4], Pintian Wang[1], Weizhen Wu[3], Hongcai Teng[4], Kai Luo[4], Xinyu Huang[4], Wenyu Feng[6], Shijie Liao [1] ✉, Juliang He [2] ✉ & Yun Liu [4] ✉

Polymorphonuclear myeloid-derived suppressor cells (PMN-MDSCs) are crucial mediators of tumor-induced immunosuppression, while their heterogeneity and spatial dynamics across malignancies remain poorly understood. By integrating single-cell RNA sequencing data from 576 samples across 19 cancer types and spatial transcriptomics data from three distinct malignancies, we identified a PMN-MDSC population. This cell population demonstrated characteristic upregulation of immunosuppressive genes and was associated with poor prognosis across multiple cancer cohorts. Notably, *TREM1* was highly expressed in PMN-MDSCs and may mediate immunosuppressive processes. Multiplex immunofluorescence demonstrated that *TREM1*+ PMN-MDSCs exhibited significantly higher distribution in tumor regions compared to non-tumor tissues. Spatial transcriptomics analysis revealed their co-localization with fibroblasts and exhausted T cells. Moreover, CellChat analysis showed that *TREM1*+ PMN-MDSCs remodeled the tumor microenvironment through interactions with diverse cellular components. Collectively, our study revealed the conserved immunosuppressive features and spatial interaction networks of *TREM1*+ PMN-MDSCs from a pan-cancer perspective, highlighting *TREM1* as a pivotal therapeutic target to disrupt PMN-MDSC-mediated tumor immune evasion.

Tumors exhibit extensive heterogeneity, involving intricate interactions between tumor cells and their surrounding microenvironment, which collectively influence tumor growth, progression and therapeutic responses[1,2]. Accumulating evidence has demonstrated that immunosuppressive cells within the tumor microenvironment (TME) contribute to establishing an immunosuppressive milieu, exerting profound impacts on tumor progression[3,4]. Notably, myeloid-derived suppressor cells (MDSCs) undergo functional and phenotypic alterations within the TME, playing a pivotal roles in regulating tumor progression and metastasis[5,6]. As a hallmark of malignancies, MDSCs drive tumor progression by promoting the formation

of pre-metastatic microenvironments and angiogenesis[4]. They also exacerbate immunosuppression through multiple mechanisms, including recruiting other immunosuppressive cells, suppressing T cell functionality and secreting reactive oxygen and nitrogen species. These processes collectively drive tumor immune evasion[4,7–9]. Consequently, MDSCs have emerged as potential therapeutic targets in cancer treatment. In humans, MDSCs are primarily categorized into polymorphonuclear myeloid-derived suppressor cells (PMN-MDSCs) and monocytic MDSCs (M-MDSCs), with PMN-MDSCs accounting for up to 80% of the total MDSC population[6]. Notably, PMN-MDSCs exhibit significant expansion across multiple cancer

[1]Department of Traumatic Orthopedic and Hand Surgery, The First Affiliated Hospital of Guangxi Medical University, Nanning, China. [2]Department of Bone and Soft Tissue Surgery, The Affiliated Tumor Hospital, Guangxi Medical University, Nanning, China. [3]Department of Medical Oncology, The First Affiliated Hospital of Guangxi Medical University, Nanning, China. [4]Department of Spine and Osteopathic Surgery, The First Affiliated Hospital of Guangxi Medical University, Nanning, China. [5]Department of Spine Surgery, The Fifth Affiliated Hospital of Guangxi Medical University, Nanning, China. [6]Department of Bone and Joint Surgery and Sports medicine, The Second Affiliated Hospital of Guangxi Medical University, Nanning, China. [7]These authors contributed equally: Yangjie Cai, Shanhang Li, Hening Li, Zhuan Zou. ✉e-mail: gxliaoshijie@163.com; hejuliang@gxmu.edu.cn; liuyun@gxmu.edu.cn

types and contribute to tumor progression through diverse mechanisms[10]. Although existing studies have elucidated the functional characteristics of PMN-MDSCs, their heterogeneity at a pan-cancer level, spatial distribution and interaction mechanisms with other cellular components within the TME remain poorly understood. Therefore, further research is crucial for deepening our understanding of these interactions and advancing the development of effective therapeutic strategies.

In recent years, the application of single-cell RNA sequencing (scRNA-seq) technology has provided a powerful tool for deciphering the heterogeneity and functional mechanisms of PMN-MDSCs. For instance, in gastric cancer, the infiltration of PMN-MDSCs and M-MDSCs has been demonstrated to contribute to the formation of an immunosuppressive TME[11]. In an scRNA-seq study of glioblastoma, Jackson et al.[12] identified a PMN-MDSC population characterized by elevated expression of *S100A8*, *S100A9*, *S100A12*, *LYZ*, and *EREG*. Although scRNA-seq enables resolution of PMN-MDSCs heterogeneity, it offers no insight into spatial information within the TME. In a spatial context, a study of prostate cancer revealed an association between club-like senescence and immunosuppressive PMN-MDSC activity. Furthermore, PMN-MDSCs could be recruited and activated by club-like cells to promote an immunosuppressive microenvironment[13]. However, current studies predominantly focus on individual cancer types, lacking systematic exploration of PMN-MDSCs functional heterogeneity and spatial distribution at a pan-cancer level. Additionally, the mechanisms through which PMN-MDSCs regulate immunosuppressive states in the TME via intercellular interactions remain to be further elucidated. The integration of single-cell transcriptomics with spatial transcriptomics (ST) not only offers promise for uncovering the spatial distribution of PMN-MDSCs, but also aids in elucidating their interactions with other cellular components in the TME. This approach provides insights for developing tumor therapeutic strategies and precision interventions.

In this study, the molecular landscape of PMN-MDSCs was systematically identified and characterized through an integrative analysis of pan-cancer scRNA-seq datasets. Subsequent analyses identified *TREM1* as a pivotal functional regulator in PMN-MDSCs, which was highly expressed in various tumor tissues, and elevated *TREM1* expression was significantly correlated with poor prognosis. T cell function was effectively inhibited by MDSCs using CCK-8 and ELISA assays, highlighting its critical involvement in mediating immunosuppressive properties. Furthermore, multiplex immunofluorescence (mIF) experiments demonstrated that *TREM1*+ PMN-MDSCs exhibited significantly higher distribution in tumor regions compared to non-tumor tissues. ST analyses further delineated the distribution of *TREM1*+ PMN-MDSCs within the TME, emphasizing their spatial co-localization with fibroblasts and exhausted T cells. Cell interaction analysis indicated that *TREM1*+ PMN-MDSCs engaged in intercellular communication via immunosuppressive signaling with stromal cells and immune cells, collectively shaping an immunosuppressive microenvironment. NicheNet analysis revealed that CD4+ Tregs and fibroblasts may jointly regulate the functional activity of *TREM1*+ PMN-MDSCs through the *LTB–TNFRSF1A* and *HAS2/CALR–CD44* signaling axes, respectively. These findings not merely advance our understanding of the functional dynamics of PMN-MDSCs, but also establish a conceptual framework for targeted immunotherapies designed to combat PMN-MDSCs-mediated immunosuppression.

## Results

### Construction of a pan-cancer single-cell transcriptomic atlas

A pan-cancer single-cell landscape was constructed by integrating data from 576 samples across 19 cancer types (Supplementary Data 1), sourced from the GEO database, along with 15 osteosarcoma samples from our own sequencing (Fig. 1A, B). The data included three sources: primary tumors (PT), normal tissues (NT) and metastatic lesions (Met). Following rigorous quality control and filtering, a total of 2,565,798 cells were retained for constructing the pan-cancer single-cell transcriptome atlas and conducting subsequent analyses. Cell populations were identified using classic marker

genes, including T/NK cell, myeloid, epithelial_1, fibroblast, osteoblast, B cell, endothelial, plasma cell, epithelial_2, lung epithelial, mast cell, osteoclast, basal cell, hepatocyte, acinar cell as well as an undefined cell population (Fig. 1C, D). Myeloid cells were extracted from the atlas, and six subpopulations were identified based on classical marker genes, including monocyte/macrophage (Mono/Macr), neutrophil, *APOE*_Macro, dendritic cell (DC), proliferative cell (Proli_cell) and plasmacytoid dendritic cell (pDC) (Fig. 1E, F). To identify the MDSC population within the myeloid cells, the shared gene signature between PMN-MDSCs and M-MDSCs, as proposed by Tsutsumi et al.[11] was applied, along with the immunosuppressive genes. The result revealed that both monocyte/macrophage and neutrophil populations exhibited high scores for the MDSC signature and immunosuppressive gene sets (Fig. 1G). Furthermore, we performed scoring analysis of myeloid cells using PMN-MDSCs signatures derived from Jiang's study[14], which revealed that neutrophils displayed the highest PMN-MDSCs signature scores (Supplementary Fig. 1A, B). These results suggested that the Mono/Macr population may contain M-MDSCs, while the neutrophil population likely contained PMN-MDSCs.

### Distinct features of neutrophil subpopulations and identification of the PMN-MDSCs population

Cluster analysis of neutrophils identified six distinct subpopulations: P0_*CXCL3*, P1_*S100A9*, P2_*TREM1*, P3_*IL1B*, P4_*IFIT2* and P5_*HSPA1B* (Fig. 2A). Among these, the P0_*CXCL3* subpopulation exhibited high expression of major histocompatibility complex (MHC) class II molecules, while the P1_*S100A9* subpopulation specifically overexpressed S100 calcium-binding protein A family genes (Supplementary Fig. 1C). Additionally, subpopulations potentially representing inflammatory responses (P2_*TREM1*) and interferon responses (P4_*IFIT2*) were identified (Supplementary Fig. 1D). ROGUE scoring revealed high consistency among the neutrophil subpopulations, indicating stable and reliable clustering results (Supplementary Fig. 1E).

To identify the PMN-MDSCs population, we applied gene signature scoring using the PMN-MDSCs-specific gene set defined by Tsutsumi et al.[11]. The results revealed that P0_*CXCL3*, P2_*TREM1*, P3_*IL1B* and P4_*IFIT2* subpopulations exhibited high PMN-MDSCs signature scores and elevated immunosuppressive gene signature scores (Fig. 2B, C). Furthermore, the P2_*TREM1* subpopulation exhibited the highest score when assessed with the PMN-MDSCs-specific gene set from Jiang et al. (Supplementary Fig. 1F)[14]. Importantly, the P2_*TREM1* subpopulation highly expressed *ITGAM*, *OLR1*, *IL1B* and *ARG1*, but low expression of *HLA-DRB1* (Fig. 2D). Notably, the P2_*TREM1* subpopulation also exhibited elevated expression of PMN-MDSCs-associated genes and immunosuppressive genes, including *CEACAM8*, *NOS2*, *ARG1*, *TGFB1*, and *VEGFA* (Fig. 2E). GSVA analysis further revealed high enrichment in pathways related to inflammation response, oxidative stress, and the negative regulation of T cell proliferation in this subpopulation (Fig. 2F). Additionally, this subpopulation demonstrated elevated signatures for chemotaxis, MDSC function, and immunosuppressive function (Fig. 2G). In a comparison of the neutrophil subpopulations from this study and data from Jiang et al.[14], the P2_*TREM1* subpopulation showed a high degree of similarity in gene expression patterns to the PMN-MDSC (Fig. 2H, Supplementary Fig. 1A). In conclusion, the P2_*TREM1* subpopulation was identified as the *TREM1*+ PMN-MDSCs population.

### Functional dynamics and molecular characteristics of *TREM1*+ PMN-MDSCs in tumor progression

To investigate the functional evolution of *TREM1*+ PMN-MDSCs during tumor progression, inter-group analyses were performed on PMN-MDSCs from NT, PT, and Met groups. Compared to the NT and Met group, the PT group demonstrated the highest enrichment scores for oxidative stress, immunosuppression, inflammatory response and epithelial-mesenchymal transition (EMT). While its antigen-presenting functionality was lower than that of the NT group (Fig. 3A). The Pathway RespOnsive GENes for activity inference (PROGENy) analysis revealed that inflammation-related

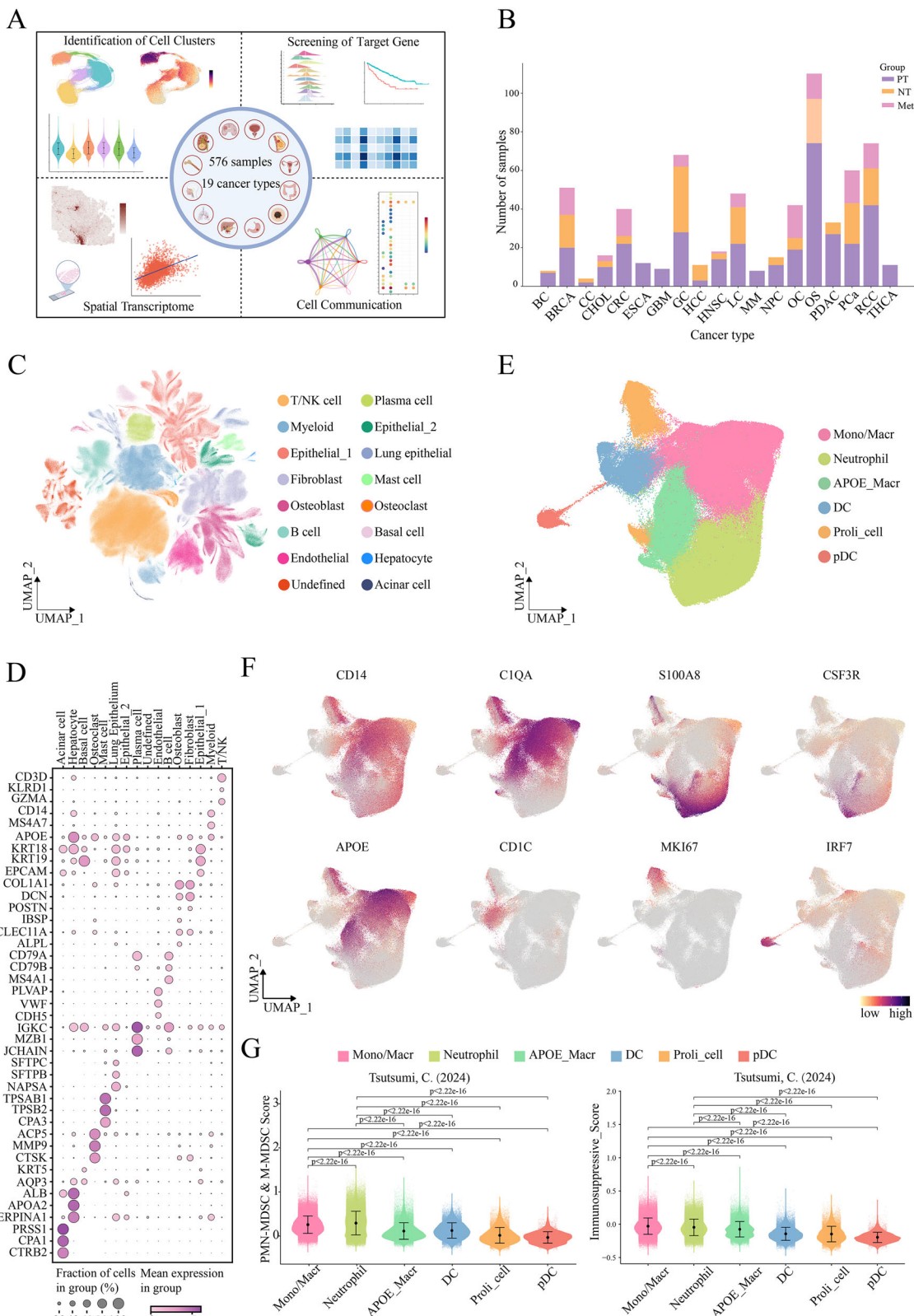

**Fig. 1 | Pan-cancer single-cell landscape. A** Workflow diagram. Created in BioRender. Bolin, L. (2025) https://BioRender.com/7l58jez. **B** Number of analysis samples in pan-cancer scRNA-seq. **C** UMAP plot showing major cell types. **D** Bubble plot displaying markers corresponding to major cell types. **E** Myeloid cell landscape. **F** Scatter plot showing markers corresponding to myeloid cell subsets. **G** Violin plot displaying gene set scores in myeloid cell subsets.

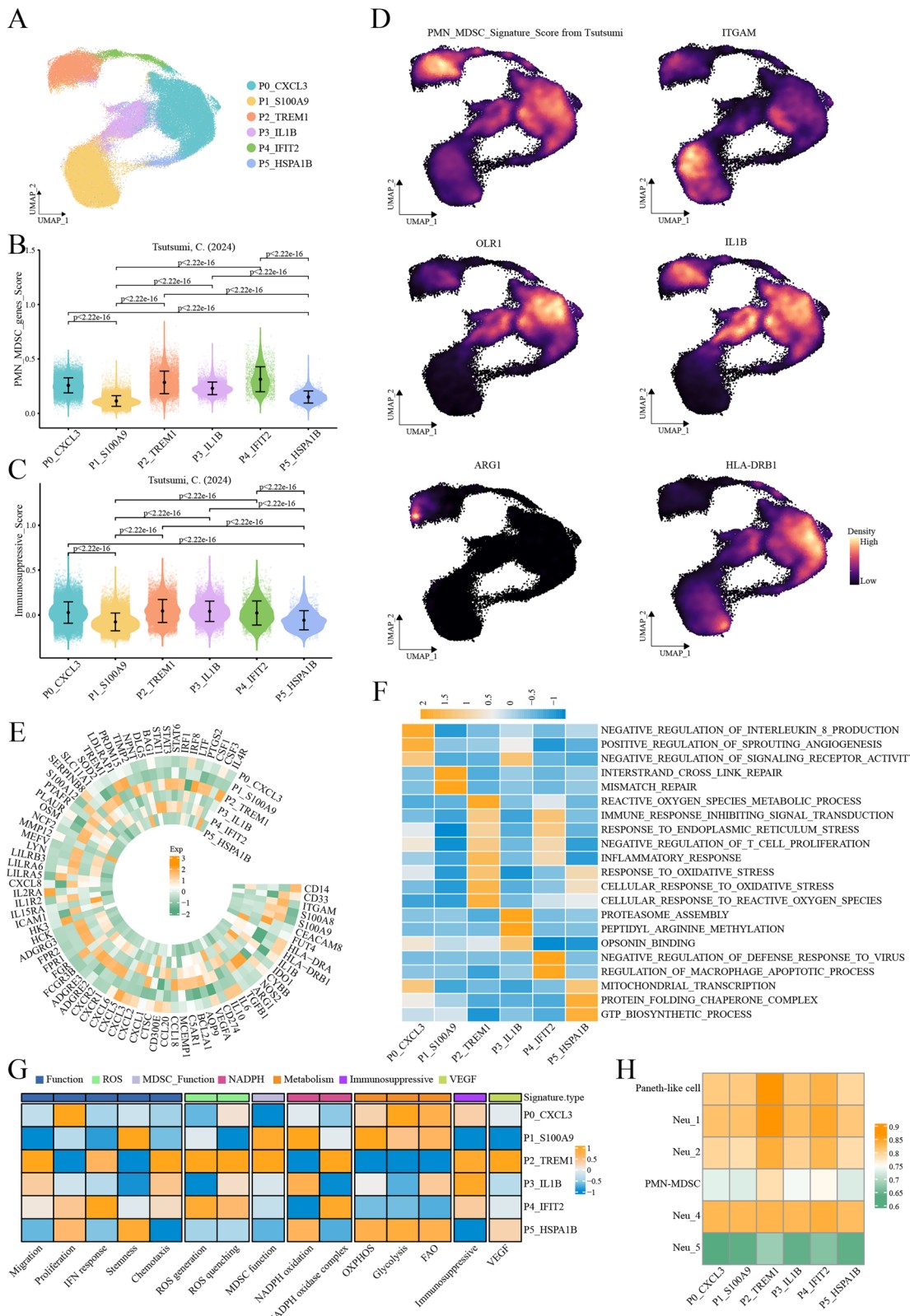

**Fig. 2 | Identification of PMN-MDSCs. A** UMAP plot showing neutrophil subsets. **B**, **C** Violin plots displaying gene set scores in neutrophil subsets, respectively showing immunosuppressive gene set and PMN-MDSCs-related gene set. **D** Density plot displaying gene sets and markers for PMN-MDSCs. **E** Circular heatmap showing gene expression of PMN-MDSCs-associated genes, immunosuppressive genes and other genes. **F** Pathway enrichment analysis in neutrophil subsets by GSVA. **G** Heatmap displaying the expression of features derived from 15 gene sets across neutrophil subsets. **H** Heatmap showing the correlation between neutrophil clusters defined by Jiang et al. through scRNA-seq and neutrophil subtypes defined by our study.

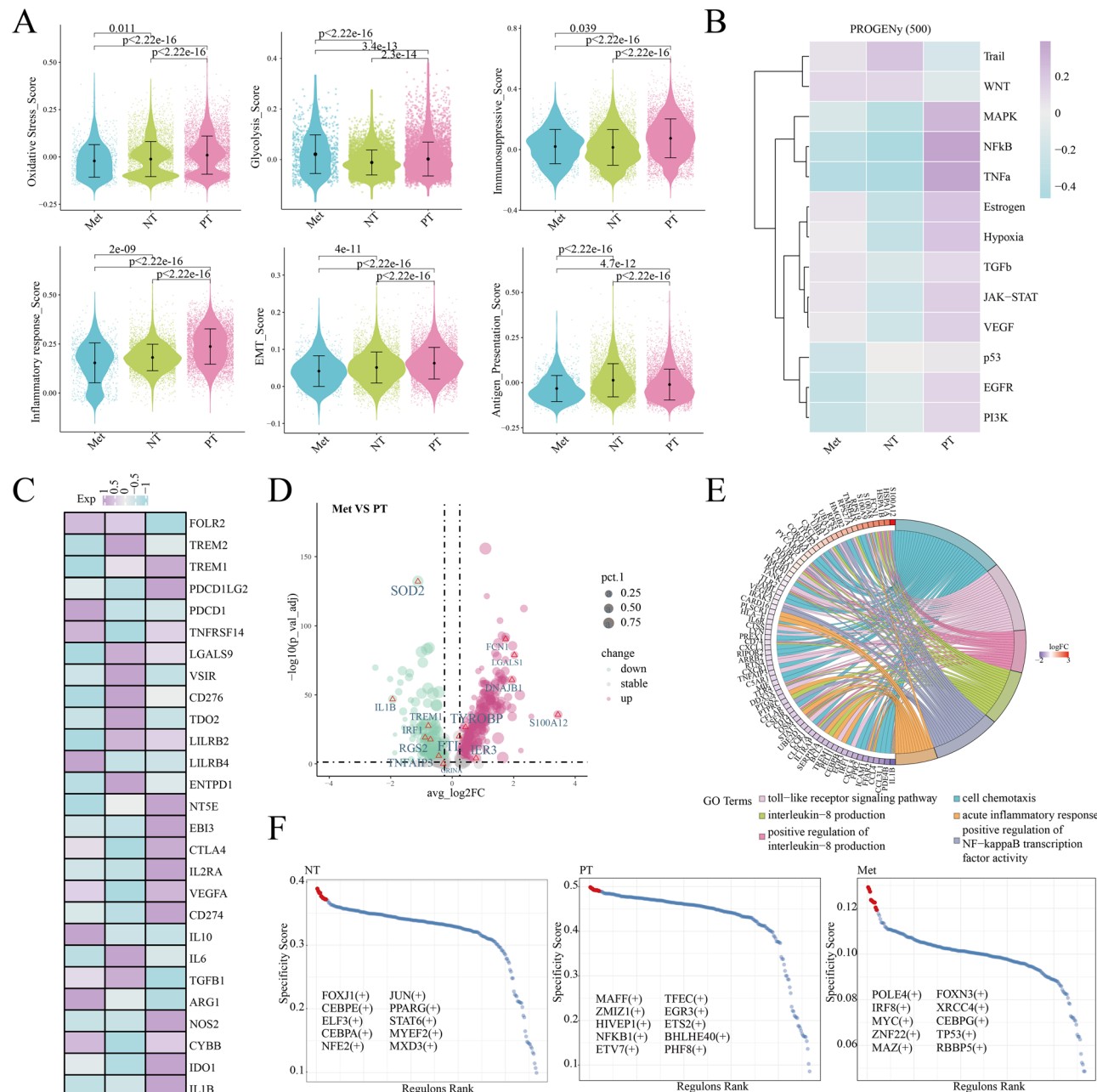

**Fig. 3 | Inter-group differential analysis of PMN-MDSCs. A** Violin plot displaying the scores of gene sets across groups. **B** Heatmap displaying the enrichment levels of signaling pathways in PMN-MDSCs across groups, as inferred by PROGENy analysis. **C** Heatmap displaying the enrichment levels of signaling pathways in PMN- MDSCs across groups, as inferred by PROGENy analysis. **D** Volcano plot showing differential genes between Met and PT groups. **E** Chord plot showing functional enrichment of differential genes between Met and PT groups. **F** Dot plot displaying the top 10 specific activated TFs ranked by regulon-specific score in each group.

pathways such as NFκB and TNFα were activited in *TREM1*[+] PMN-MDSCs in the PT group (Fig. 3B). Besides, we found high expression of immunosuppressive genes including *IDO1*, *NOS2*, and *TREM1* within this group (Fig. 3C). Compared to the Met group, PMN-MDSCs in the PT group exhibited higher expression of inflammation and immunosuppression related genes, including *IRF1*, *IL1B* and *TREM1*[15–17]. While genes associated with angiogenesis and inflammatory responses were upregulated in the Met group, such as *LGALS1* and *TYROBP* (Fig. 3D)[18,19]. Enrichment analysis using clusterProfiler revealed significant enrichment of differentially expressed genes (DEGs) in pathways such as the positive regulation of NF-κB transcription factor activity and the acute inflammatory response (Fig. 3E). These results suggested that *TREM1*[+] PMN-MDSCs exhibited stage-specific functional dynamics during tumor progression. To identify

active transcription factors (TFs) at each stage, pySCENIC analysis was performed. Specifically, we noticed that TFs associated with tumor invasion, progression and immunosuppressive functions, such as *MAFF*, *NFKB1*, and *ETV7*, exhibited higher activity in the PT group[20–22]. By contrast, in the Met group, TFs related to tumor angiogenesis, metastasis and EMT, such as *XRCC4*, *MYC* and *MAZ* showed elevated activity (Fig. 3G)[23–25]. These findings suggested that distinct TFs regulate PMN-MDSCs at different stages to modulate their functions.

**TREM1 was a key driver of immunosuppression in PMN-MDSCs**
To identify the target genes in *TREM1*[+] PMN-MDSCs, an intersection analysis was performed between the DEGs upregulated in this population and previously reported PMN-MDSCs marker genes[26,27]. This analysis

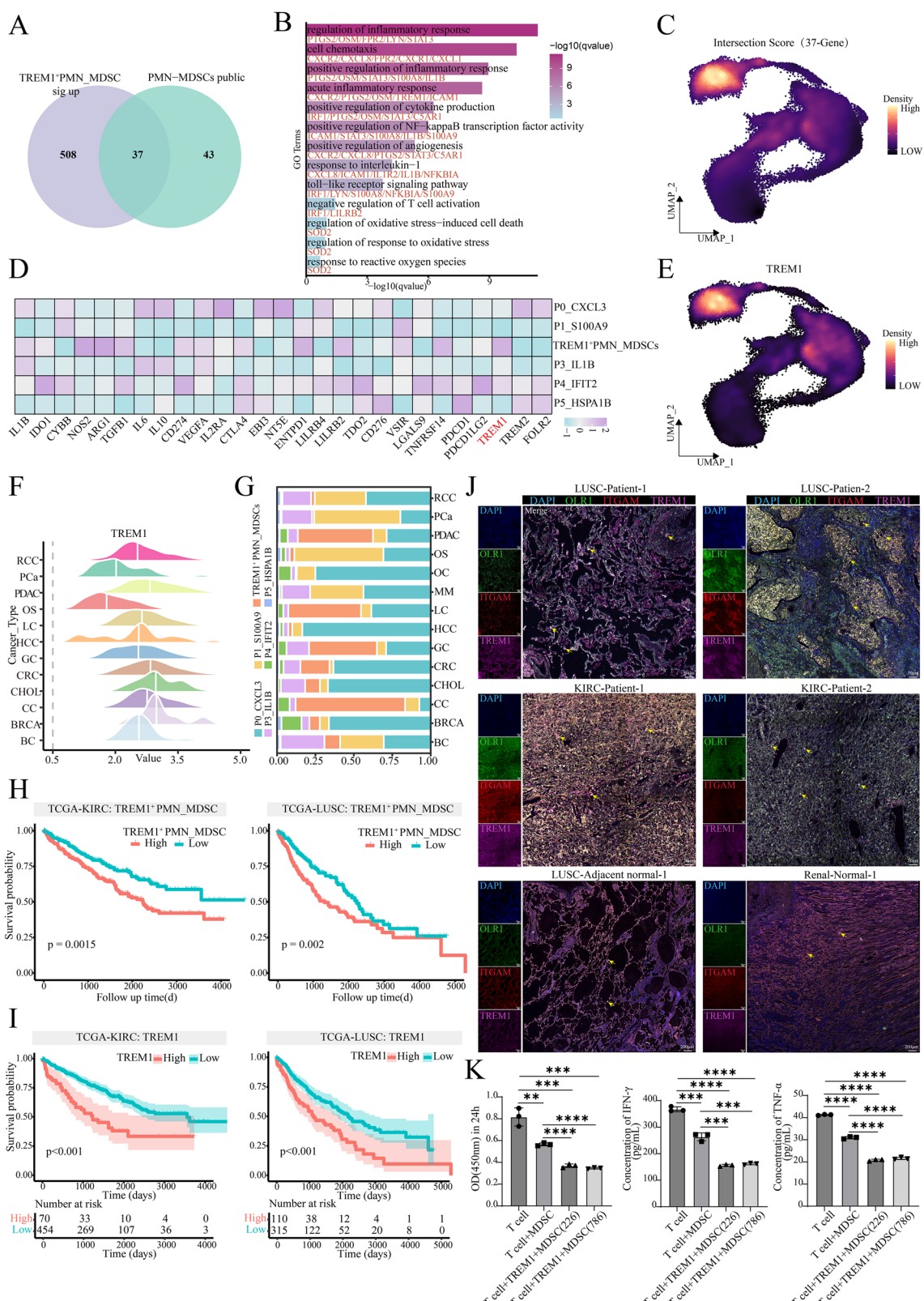

resulted in the identification of 37 shared genes (Fig. 4A, Supplementary Data 2). To further investigate the functions of these genes, functional enrichment analysis was performed using clusterProfiler. The result indicated that inflammatory pathways (regulation of inflammatory response, acute inflammatory response and positive regulation of NF-κB transcription factor activity) and angiogenesis pathway were activated (Fig. 4B). The

expression distribution of the shared genes was further visualized using a UMAP plot (Fig. 4C). To explore the potential immunosuppressive role of *TREM1*⁺ PMN-MDSCs, the expression of immunosuppressive genes was assessed. Among these, triggering receptor expressed on myeloid cells 1 (*TREM1*), an immunosuppressive gene[17], was found to be highly expressed in the tumor-associated PMN-MDSCs population (Fig. 3C, Fig. 4D, E). In

**Fig. 4 | *TREM1* was a key driver of immunosuppression in PMN-MDSCs. A** Venn diagram showing the overlap of upregulated differential genes in the *TREM1*⁺ **PMN-MDSCs** subset and genes reported in the literature for PMN-MDSCs. **B** Bar plot displaying the enriched pathways for the intersecting genes. **C** Density plot showing the expression of the intersecting genes. **D** Heatmap showing the expression of immunosuppressive genes in neutrophil subsets. **E** Density plot showing the expression of the *TREM1* gene. **F** The ridge plot displaying the expression of *TREM1* in the PMN-MDSCs subgroup across various cancer types that contained ≥ 10 cells in this subgroup. **G** Proportional plot showing the distribution of neutrophil subsets across different cancer types. **H** Survival analysis displaying the prognosis of

*TREM1*⁺ PMN-MDSCs. **I** Survival analysis displaying the prognosis of the *TREM1* gene. **J** Multiplex immunofluorescence staining for *TREM1*⁺ PMN-MDSCs. DAPI (blue), *ITGAM* (red), *OLR1* (green) and *TREM1* (magenta) are shown in individual and merged channels. The yellow arrows point to cells positive for the three markers (LUSC, lung squamous cell carcinoma; KIRC, kidney renal clear cell carcinoma). **K** Compared with the T cell group, the function of T cells was inhibited after co-culture with MDSCs. When MDSCs co-cultured with tumor cells were added to T cells, the inhibition of T cell function became more significant. The statistical method used was an unpaired *t*-test. Error bars represent the standard error (SE). **p* < 0.05, ***p* < 0.01, ****p* < 0.001, *****p* < 0.0001.

addition, analysis of the *TREM1*⁺ PMN-MDSCs population revealed that *TREM1* exhibited high expression in multiple cancers, including renal cell carcinoma (RCC), pancreatic ductal adenocarcinoma (PDAC), gastric cancer (GC) and lung cancers (LC) (Fig. 4F). Furthermore, the proportion of *TREM1*⁺ PMN-MDSCs also varied markedly among these cancer types (Fig. 4G). To further validate these findings, TCGA pan-cancer box was plotted by https://www.bioinformatics.com.cn (last accessed on 1 August 2025), an online platform for data analysis and visualization[28]. The results revealed that *TREM1* was highly expressed in tumor samples across multiple cancer types (Supplementary Fig. 2A). Correlation analysis of bulk RNA-seq data from TCGA database further confirmed that this gene was associated with immunosuppression, such as in lung squamous cell carcinoma (LUSC), kidney clear cell carcinoma (KIRC), low-grade glioma (LGG) and glioblastoma (GBM) (Supplementary Fig. 2B).

To further explore the function of *TREM1* gene, PMN-MDSCs were divided into *TREM1*-high group and *TREM1*-low group. The former was found to be significantly enriched in chemotaxis and response to oxidative stress pathways, while the latter was enriched in cytoplasmic translation and ribosome biogenesis pathways (Supplementary Fig. 2C). Subsequently, the prognosis analysis revealed that patients with high infiltration of *TREM1*⁺ PMN-MDSCs, such as those with KIRC, LUSC and LGG, exhibited a significantly poor prognosis. However, in skin cutaneous melanoma (SKCM), the high-infiltration group showed a better prognosis than the low-infiltration group (Fig. 4H, Supplementary Fig. 2D). Similarly, high *TREM1* expression was found to be associated with poor prognosis in multiple cancer types, including KIRC, LUSC and LGG (Fig. 4I, Supplementary Fig. 2E). Immune cell infiltration levels were further assessed using the CIBERSORT algorithm. In the TCGA-KIRC and TCGA-LUSC cohorts, infiltration levels of various immune cells, including naive B cells, plasma cells, CD8⁺ T cells and activated NK cells, were significantly reduced in the *TREM1*-high group (Supplementary Fig. 2F). These findings suggested that high *TREM1* expression was associated with the formation of an immunosuppressive microenvironment, leading to reduced infiltration of anti-tumor immune cells. Next, we performed mIF staining, which confirmed the presence of *TREM1*⁺ PMN-MDSCs in LUSC and KIRC. In contrast, these cells were significantly reduced in the adjacent normal tissue of LUSC and normal renal tissue, with markedly lower *TREM1* expression levels compared to tumor tissue (Fig. 4J, Supplementary Fig. 3A). To validate the impact of MDSCs on T cell function, MDSCs isolated from healthy mice spleens were co-cultured with T cells and tumor cells. We found that co-culturing MDSC with T cells inhibited the activity of T cells and significantly reduced the secretion levels of IFN-γ and TNF-α effector molecules. Notably, co-culture with *TREM1*⁺ MDSCs and tumor cells resulted in more pronounced inhibition of T cell function, confirming that MDSCs exerted their suppressive effects via *TREM1* (Fig. 4K).

## Association of high infiltration of *TREM1*⁺ PMN-MDSCs with immunosuppression, EMT, fibroblast, and exhausted T cell across multiple cancer types

To further explore the association between *TREM1*⁺ PMN-MDSCs and cell populations or key biological processes in the TME across different cancer types, a systematic analysis was conducted using TCGA data. The results revealed significant positive correlations between *TREM1*⁺ PMN-MDSCs scores and immunosuppression, exhausted T (Tex) cells, fibroblasts and

EMT scores (Fig. 5A). Notably, in three representative cancer types—KIRC, LUSC and LGG—the high *TREM1*⁺ PMN-MDSCs infiltration group exhibited significantly higher scores for these features compared to the low-infiltration group. Moreover, the scores of *TREM1*⁺ PMN-MDSCs were positively correlated with gene set scores for these features (Fig. 5B, C). scRNA-seq data analysis showed that high-infiltrating *TREM1*⁺ PMN-MDSCs also exhibited higher immunosuppression and EMT scores (Supplementary Fig. 3B). Based on these results, we hypothesized that *TREM1*⁺ PMN-MDSCs, a cell population associated with immunosuppression, may be closely linked to the infiltration of fibroblasts, Tex, and the EMT process in the TME through complex cell-cell interactions and signaling pathways.

## Spatial transcriptomics revealed the association of *TREM1*⁺ PMN-MDSCs with fibroblast, Tex cell and immunosuppressive features

To further investigate the spatial distribution of *TREM1*⁺ PMN-MDSCs, ST analysis was conducted using specific gene sets (Supplementary Data 2). In tumor tissue sections from breast cancer (BRCA), LC, and KIRC, regions containing *TREM1*⁺ PMN-MDSCs exhibited significantly enhanced immunosuppressive activity, suggesting a role for *TREM1*⁺ PMN-MDSCs in promoting immunosuppression within the TME (Fig. 6A). Additionally, the regions of *TREM1*⁺ PMN-MDSCs infiltration were found to be adjacent to fibroblasts, with enhanced EMT activity (Fig. 6A, Supplementary Fig. 4A). Bayesian-enhanced spatial data highlighted that regions with high *OLR1* expression also had increased *TREM1* and *COL1A1* expression (Fig. 6B). In KIRC, *TREM1*⁺ PMN-MDSCs were found enriched in regions surrounding infiltrating epithelial/tumor cells and exhibited spatial co-localization with Tex cells and fibroblasts (Fig. 6C, Supplementary Fig. 4B). Critically, *TREM1*⁺ PMN-MDSCs-high spots showed increased infiltration of Tex (Fig. 6D). Further analysis found that significant positive correlations between *TREM1*⁺ PMN-MDSCs levels and both fibroblast and Tex (Fig. 6E, F). These results demonstrated that *TREM1*⁺ PMN-MDSCs exerted immunosuppressive effects within the TME and may also modulate it through spatial interactions with fibroblasts and Tex cells.

## *TREM1*⁺ PMN-MDSCs promoted an immunosuppressive TME through multicellular interactions

To investigate the interactions between *TREM1*⁺ PMN-MDSCs and other cell types, we first incorporated stromal and immune cells and assessed the characteristic scores of immunosuppressive gene sets. The data revealed that *TREM1*⁺ PMN-MDSCs exhibited the highest scores, followed by regulatory T cells (*CD4*⁺ Treg) (Supplementary Fig. 5A–C). CellChat analysis showed a close interaction between fibroblasts and *TREM1*⁺ PMN-MDSCs (Fig. 7A). As shown in Fig. 7B, fibroblasts and endothelial cells primarily function as signal senders, exhibiting a strong outgoing interaction strength, while effector T cells (*CD8*⁺ Teff) mainly act as receivers, showing a close incoming interaction strength.

Our analysis demonstrated that MHC-I and MIF signaling were widely activated in the TME, indicating their broad role within the TME. These signals were sent by various cell types and primarily received by *CD8*⁺ Teff (Fig. 7C, Supplementary Fig. 6A). Among these, the ligand-receptor pairs, such as *HLA-E - KLRK1* and *MIF - (CD74 + CXCR4)*, contributed most significantly to the signaling pathways. Notably, *TREM1*⁺ PMN-MDSCs were also involved in pro-tumor signaling pathways, such as *FN1* and

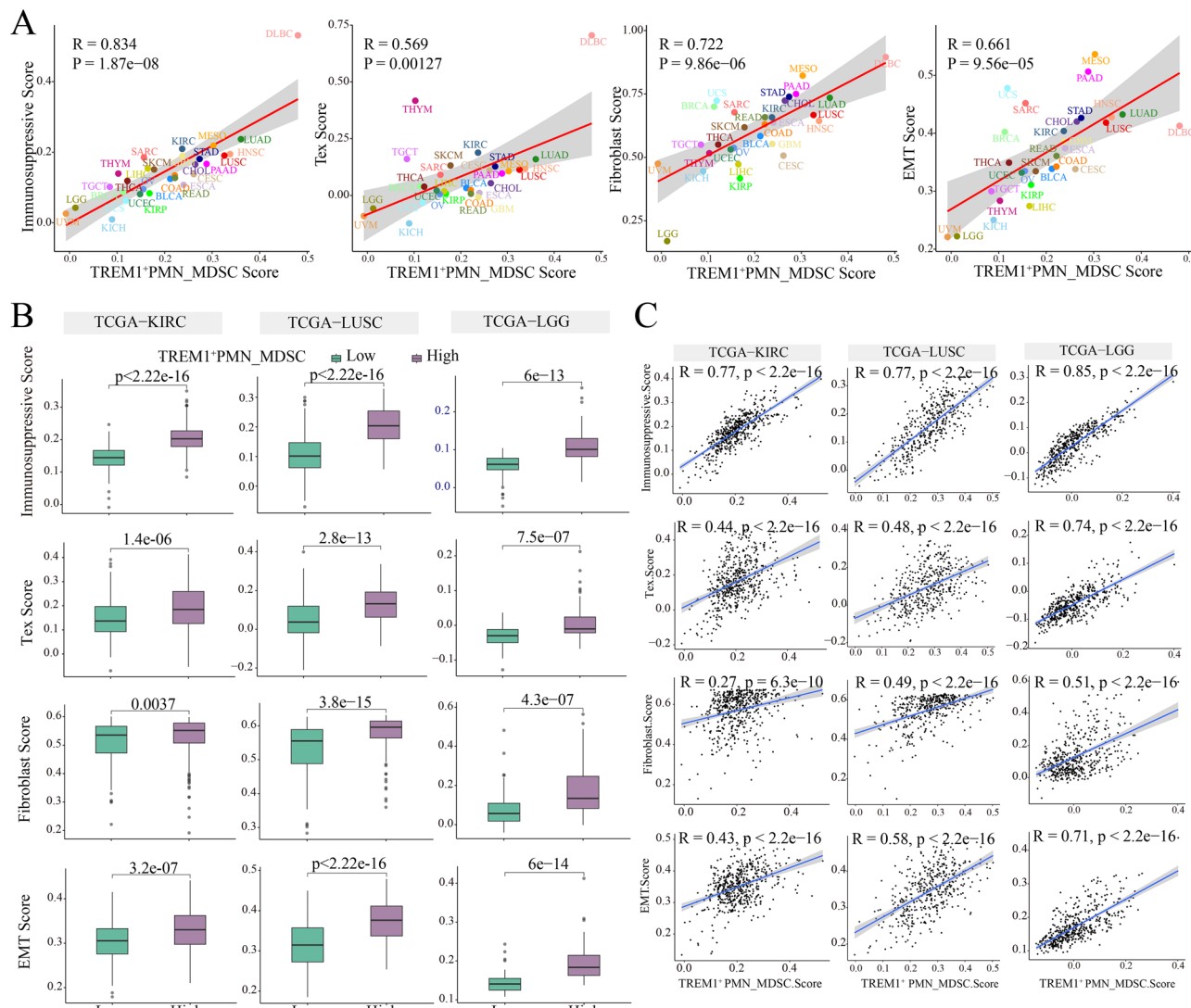

**Fig. 5 | Association of *TREM1*+ PMN-MDSCs with immunosuppression, exhausted T cells, EMT and fibroblasts. A** Scatter plot showing the correlation between *TREM1*+ PMN-MDSCs scores and gene sets across different cancer types.

**B** Box plot displaying the scores of gene sets in high and low infiltration groups of *TREM1*+ PMN-MDSCs in bulk RNA-seq. **C** Scatter plot showing the correlation between *TREM1*+ PMN-MDSCs and other gene sets in bulk RNA-seq.

*ANNEXIN*. In the *FN1* signaling pathway, the *FN1 - CD44* ligand-receptor interaction contributed most significantly, while in the *ANNEXIN* signaling pathway, the *ANXA1 - FPR1* ligand-receptor pair was notably active (Supplementary Fig. 6B, C). Further analysis revealed that endothelial cells, fibroblasts, CD8+ Teff, and NK cells significantly increased the communication with *TREM1*+ PMN-MDSCs via immunosuppressive pathways *ANXA1 - FPR2* and *ANXA1 - FPR1*[29]. In addition, we found that fibroblasts, as signal senders, may cooperate with *TREM1*+ PMN-MDSCs in extracellular matrix remodeling and tumor progression by secreting collagen and *FN1* to interact with the *CD44* (Fig. 7D). Using the stLearn method, the spatial interactions of the *ANXA1 - FPR1* and *FN1 - CD44* ligand-receptor pairs were validated (Fig. 7E). When *TREM1*+ PMN-MDSCs acts as a signal senders, they had close interaction with CD8+ Teff and endothelial cells (Supplementary Fig. 6D). Subsequently, NicheNet was performed to infer potential interactions between ligands, receptors and targets genes across these cell clusters. A heatmap ranked by AUPR values showed that *TNF* exhibited the highest activity (Fig. 7F). As shown in Fig. 7G, *CD4*+ Tregs highly expressed the *TNF* and *LTB* ligand, while fibroblasts expressed ligands such as *HAS2*, *VEGFA*, *CXCL12*, *CALR*, *COL18A1* and *MIF*. We noted that *TNF* exhibited high regulatory potential over *TREM1*+ PMN-MDSCs. Among all identified ligands, the *LTB*, *HAS2* and *CALR* ligands

had the highest regulatory relationship with the *TREM1* gene (Fig. 7H). Meanwhile, analysis of receptors for these highly active ligands identified *CD44* as the high-potential receptor for *CALR* and *HAS2*, and *TNFRSF1A* for *LTB* (Supplementary Fig. 6E). These results indicated that CD4+ Tregs and fibroblasts may jointly regulate the functional activity of *TREM1*+ PMN-MDSCs through the *LTB–TNFRSF1A* and *HAS2/CALR–CD44* signaling axes, respectively.

## Discussion

Immunosuppressive cells in the TME can be induced or recruited by tumors, contributing to the establishment of an immunosuppressive environment and promoting tumor immune escape[30]. MDSCs play a key role in these processes[8]. PMN-MDSCs, a major component of MDSCs, also participate in tumor progression through multiple pathways[31]. However, their heterogeneity at the pan-cancer level, the spatial distribution, and the mechanisms of interaction with other cellular components in the TME remain poorly understood. In this study, a pan-cancer single-cell transcriptomic map was constructed based on scRNA-seq data from multiple cancer types. A population of *TREM1*+ PMN-MDSCs was identified. Then, the functional characteristics of these cells and their interactions with other cell subpopulations were analyzed. Our research revealed the molecular

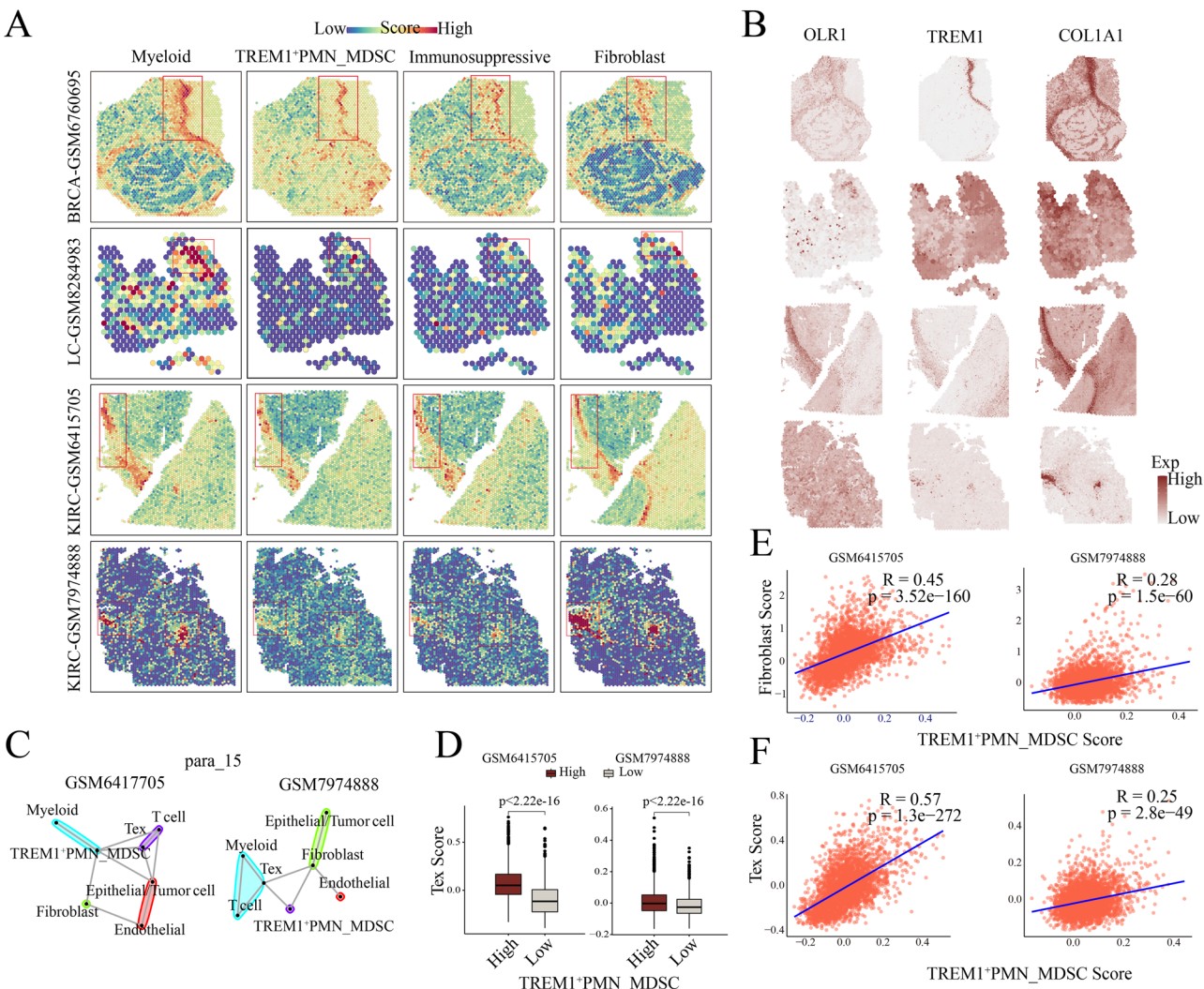

**Fig. 6 | Spatial distribution characteristics of *TREM1*⁺ PMN-MDSCs. A** Spatial feature plots of myeloid cells, *TREM1*⁺ PMN-MDSCs, fibroblasts, along with the immunosuppressive score, in tissue sections from breast cancer (BRCA), lung cancer (LC) and kidney clear cell carcinoma (KIRC). **B** Enhanced spatial feature plots showing the expression of *OLR1*, *TREM1* and *COL1A1* in tumor tissues. **C** MistyR analysis showing the co-localization of *TREM1*⁺ PMN-MDSC with other cell types in KIRC. **D** Box plot showing the expression of exhausted T cells between high and low infiltration groups of *TREM1*⁺ PMN-MDSCs in KIRC. **E** Scatter plot showing the correlation between *TREM1*⁺ PMN-MDSCs and fibroblasts in KIRC. **F** Scatter plot showing the correlation between *TREM1*⁺ PMN-MDSCs and exhausted T cells in KIRC.

conservation of *TREM1*⁺ PMN-MDSCs population across various cancers and their clinical prognostic value. The existence of these cells and their immunosuppressive function were confirmed using clinical samples and in vitro experiments.

MDSCs result from the abnormal differentiation of myeloid cells under pathological conditions and possess potent immunosuppressive activity. These cells are primarily composed of PMN-MDSCs, derived from the granulocytic lineages, and M-MDSCs, originating from the monocytic lineages[32]. In most cancer, PMN-MDSCs are the most abundant MDSC population. These cells effectively suppress T cell activity within tumors and promote immune evasion through mechanisms such as upregulation of arginase 1 (*ARG1*), nitric oxide synthase 2 (*NOS2*), and reactive oxygen species production[26,31,33]. He et al.[31] found that in colorectal cancer, the lactate receptor *HCAR1* inhibited CD8⁺ T cell-mediated antitumor immunity by promoting *CCR2*⁺ PMN-MDSC recruitment. Wang et al.[10] discovered that *CD300ld*, a surface marker of PMN-MDSC, regulated the downstream *S100A8/A9* axis via *STAT3* signaling, promoting PMN-MDSC migration and enhancing T cell suppression. Our study mainly focused on the pan-cancer level. According to the specific PMN-MDSCs gene set, the *TREM1*⁺ PMN-MDSCs population with immunosuppressive function was

identified. Meanwhile, the functional characteristics of this population at different stages of tumor progression were revealed. Our study confirmed that these cells exhibited significant immunosuppressive characteristics across various cancers, with high expression of key genes such as *TGFB1*, *TREM1* and *STAT3*. These cells were also significantly enriched in pathways related to the negative regulation of T cell proliferation, oxidative stress and endoplasmic reticulum stress. These findings were consistent with existing studies and supported the central role of PMN-MDSCs in tumor immune evasion. Subsequently, the spatial distribution of cells was analyzed using the ST data. We observed that *TREM1*⁺ PMN-MDSCs co-localized with fibroblasts and their infiltration regions showed upregulation of immuno-suppressive and EMT signatures. In addition, in KIRC, we also observed infiltration of exhausted T cells in the region rich in *TREM1*⁺ PMN-MDSCs, suggesting that these cells may induce T cell exhaustion.

Triggering receptor expressed on myeloid cells 1 (*TREM1*), a pro-inflammatory receptor, is widely expressed on myeloid cells. It not only amplifies inflammatory responses by mediating pro-inflammatory signaling but also promotes tumor progression, immune evasion, and resistance by activating tumor-infiltrating myeloid cells[34,35]. Previous studies showed that tumor growth in murine melanoma and fibrosarcoma models could be

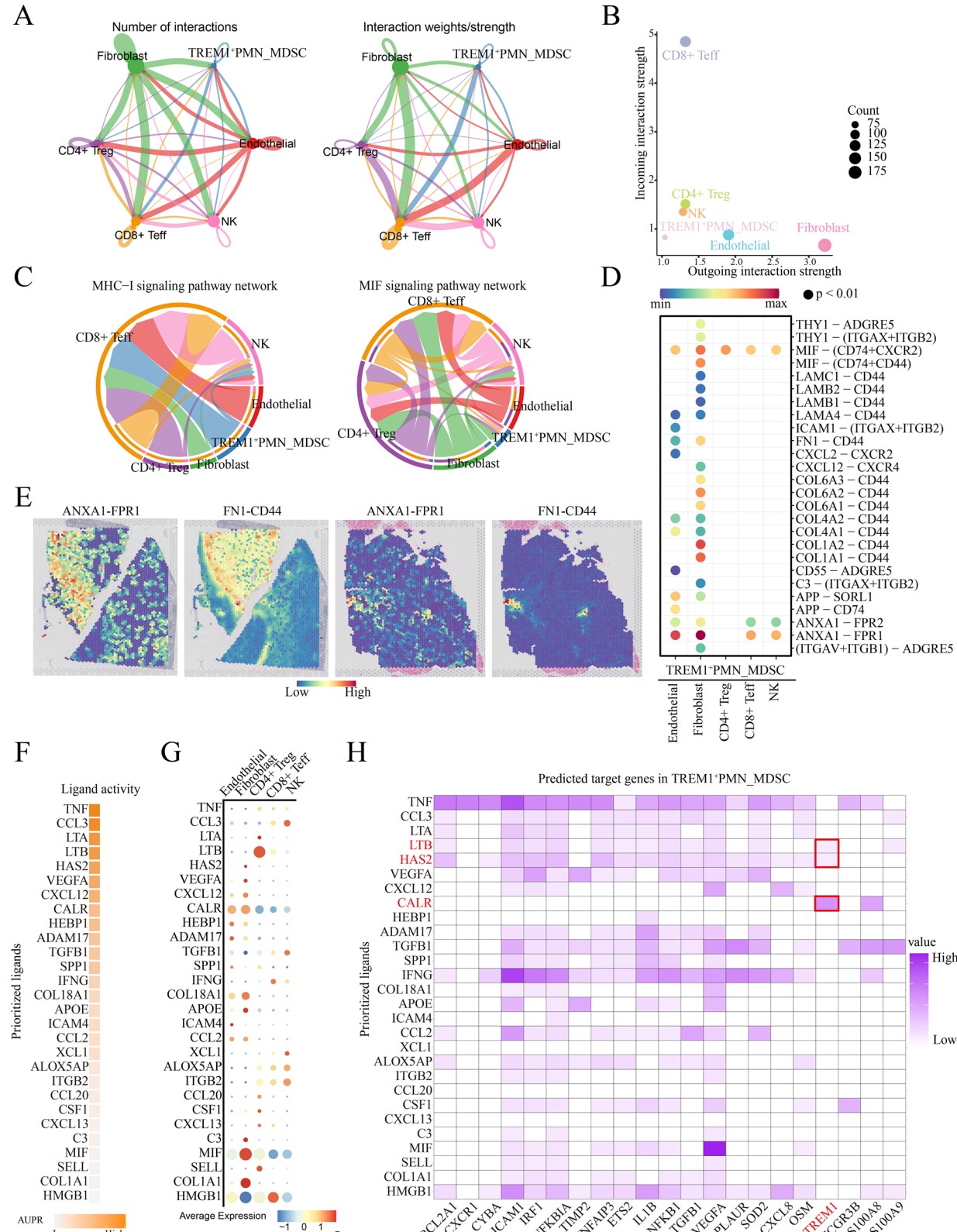

**Fig. 7 | *TREM1*⁺ PMN-MDSCs promote an immunosuppressive tumor micro-environment through multicellular interactions. A** The cell-cell interactions networks. **B** Scatter plots showing the strength of signal outgoing and incoming by different cell populations. **C** Chord plot showing the MHC-I and MIF signaling pathway networks. **D** Bubble plot showing ligand-receptor pairs of *TREM1*⁺ PMN-MDSCs as receivers interacting with other cell types. **E** Spatial feature plots showing the interaction activity of selected ligand-receptor pairs in KIRC tissue sections. **F** Top-ranked ligands inferred to regulate *TREM1*⁺ PMN-MDSC svia endothelial cells, fibroblasts, CD4⁺ Tregs, CD8⁺ Teffs and NK cells according to NicheNet. **G** Dot plot showing the expression percentage (dot size) and intensity (dot intensity) of top-ranked ligands in different cell types. **H** Heatmap depicting the regulatory relationships between ligands and target genes.

delayed by inhibiting *TREM1* gene expression. Additionally, knockout of the *TREM1* gene reduced the immunosuppressive function of MDSCs, along with an increase in the cytotoxic CD8$^+$ T cell population and upregulation of PD-1 expression[17]. Another study demonstrated that the *TREM1* agonist PY159 activated tumor myeloid cells, promoting a pro-inflammatory TME[34]. Consistent with previous studies[17], our study found that *TREM1* was highly expressed in tumor-infiltrating PMN-MDSCs, and this expression correlated with immunosuppressive function and poor prognosis. Importantly, our experiment confirmed that *TREM1*$^+$ PMN-MDSCs were induced and expanded within tumor tissues. Collectively, *TREM1* acted as a pivotal immunosuppressive gene in this cell population. Targeting *TREM1* could disrupt the immune evasion mediated by these cells.

Intercellular communication is essential for maintaining the function of organisms. MDSC has been proven to interact with other immune cells to inhibit antitumor activity[36]. Early studies also found that MDSC can present antigens to CD8$^+$ T cells through MHC-I molecules, thereby inhibiting their activation[37]. Similar to the results found in this literature, our findings indicated that when *TREM1*$^+$ PMN-MDSCs acted as the sender, there was a close communication mediated by MHC-I molecules between it and CD8$^+$ Teff cells, which might be crucial for the formation of the immunosuppressive microenvironment. More importantly, our cell communication analysis revealed that fibroblasts and CD4$^+$ Tregs regulated *TREM1*$^+$ PMN-MDSCs via the *HAS2/CALR-CD44* and *LTB-TNFRSF1A* axes, respectively. These results provide insights for determining therapeutic strategies targeting PMN-MDSCs.

Our study also had certain limitations. First, although 19 cancer types were included, the sample size for certain cancers was limited, and future studies will need to expand the cohort to validate the generalizability of the cell population characteristics. Additionally, while ST provided us with spatial distribution characteristics of multiple cellular components, the specific molecular mechanisms underlying their interactions still require elucidation through functional experiments.

In conclusion, our study integrated pan-cancer scRNA-seq and ST data to identify PMN-MDSCs with significant immunosuppressive features, revealing the molecular conservation of this cell population across various cancers. Additionally, we also analyzed the interactions between these cells and various cellular components in the TME, providing insights into the formation of the immunosuppressive microenvironment. Importantly, our findings also demonstrated that *TREM1* could serve as a potential target for cancer immunotherapy. This study provided a theoretical foundation for developing immunotherapeutic strategies targeting PMN-MDSCs.

## Methods

### Data collection and generation
A total of 576 single-cell RNA sequencing datasets were integrated from the Gene Expression Omnibus databases (GEO) (https://www.ncbi.nlm.nih.gov/geo/) and the Genome Sequence Archive (GSA) of the National Genomics Data Center (https://ngdc.cncb.ac.cn/gsa-human), including 15 osteosarcoma samples from our previous studies (Supplementary Data 1). Osteosarcoma samples obtained after surgical resection were immediately preserved on ice, minced into 1 mm³ tissue pieces, and then processed into single-cell suspensions. Single-cell transcriptome amplification and library construction were performed according to standard protocols. Finally, the single-cell libraries were sequenced on the Illumina HiSeq X Ten platform. Additionally, bulk RNA-seq data and clinical information from TCGA database (https://cancergenome.nih.gov/) were incorporated. Four spatial transcriptome sequencing samples were also obtained from the GEO database (Supplementary Data 3). The full names of all cancer types can be found in Supplementary Data 4.

### Inclusion criteria for scRNA-seq data
To minimize batch effects arising from different experimental protocols, our study incorporated scRNA-seq datasets generated exclusively using the 10x Genomics platform. The samples encompassed 19 tumor types, comprising

170 normal samples, 308 primary tumor samples and 98 metastatic samples. For the normal group, both non-tumor tissues from cancer patients and tissues from healthy individuals were included.

### scRNA-seq data analysis
The genes in the datasets were realigned according to the GRCh38 human reference genome. Quality control was performed, excluding cells with fewer than 200 expressed genes, those with fewer than 3 cells expressing each gene, or cells with more than 10% mitochondrial gene content. The cell-gene count matrix was loaded and analyzed using the Scanpy Python package (v.1.10.1). Batch effects were corrected using the SCVI model. Doublets were identified using the Scrublet algorithm, and any entries flagged as doublets were removed. Dimensionality reduction was then performed using the Scanpy toolkit. The cells were clustered at a resolution of 0.1, with the resulting populations annotated according to the cell marker genes. Subsequently, myeloid cells were isolated for further analysis. Within the R environment, dimensionality reduction and clustering were performed with a resolution of 0.1. To ensure the accuracy of the clustering results, T/NK cells and myofibroblasts were excluded from the myeloid cells. Subsequently, dimensionality reduction and clustering (resolution = 0.1) process on the purified cells was repeated. Ultimately, six myeloid cell subpopulations were obtained for analyses. Similarly, the T/NK cells were extracted and clustered by *sc.tl.leiden* function at a resolution of 0.5. After excluding the hybrid subgroups, the cell types were annotated according to the marker genes for subsequent analysis.

### Differential gene expression analysis
To identify DEGs between various neutrophil subpopulations, we utilized the scProgram package (v0.0.0.9) to select the characteristic genes for each cell subpopulation. A heatmap of these characteristic genes was then plotted using the *HeatFeatures* function. Furthermore, to examine the differential expression of *TREM1*$^+$ PMN-MDSCs between Met and PT groups, the *FindMarkers* function was employed. Genes with p_val_adj < 0.05 and avg_log2FC < −0.25 were considered down-regulated, while those with p_val_adj < 0.05 and avg_log2FC > 0.25 were considered up-regulated. To assess the effect of *TREM1* on PMN-MDSCs, these cells were categorized into *TREM1*-high and *TREM1*-low groups with the 70th percentile of *TREM1* expression serving as the grouping threshold. The *FindMarkers* function was then used to calculate DEGs between these two groups.

### Functional enrichment analysis
To explore the functional characteristics of neutrophil subpopulations, the scProgram package (v0.0.0.9) was used to quantify and visualize transcriptional programs at the single-cell level, utilizing the HALLMARK gene set for enrichment analysis. The GSVA package (v1.42.0) was also employed to assess gene set enrichment results for neutrophil subpopulations. The *enrichGO* function from the clusterProfiler (v4.2.2) package was applied for enrichment analysis on DEGs between Met and PT groups, with thresholds set at pvalueCutoff of 0.05 and qvalueCutoff of 0.05.

### scRNA-seq gene set scoring analysis
Gene set signature scores were calculated using the *AddModuleScore* function implemented in the Seurat package. The gene sets for each signature were detailed in Supplementary Data 5. Inter-group differences in signature scores were statistically assessed using the Wilcoxon rank-sum test.

### Definition of marker gene set
To assist in the identification of cell population from single-cell transcriptomic data, the MDSC gene sets provided by Tsutsumi et al.[11] were utilized to perform cell characteristic scoring. Additionally, gene sets related to PMN-MDSCs, immune suppression functions, EMT and exhausted T cells were defined. Specifically, the PMN-MDSCs gene signature was derived by intersecting the upregulated DEGs of the *TREM1*$^+$ PMN-MDSCs (i.e., P2_*TREM1* subset) identified in this study with the PMN-

MDSCs gene set reported in the literature. Other gene sets used were detailed in the Supplementary Data 2.

## Transcription factor regulon analysis

The activated TFs of $TREM1^+$ PMN-MDSCs in each group were identified using pySCENIC (v0.12.1). Among them, the regulon specificity scores (RSS) were calculated using the *calcRSS* function. The top TFs from each group were then visualized using rank-ordered scatter plots.

## TCGA RNA-seq data analysis

Normalized RNA-seq expression data and clinical information for 29 cancer types were downloaded from the TCGA database using the TCGAbiolinks package. To assess the prognostic value of $TREM1^+$ PMN-MDSCs and specific genes across different cancer types, survival analysis was performed using TCGA RNA-seq data. Based on the feature scores calculated from the intersected gene set, samples were classified into high and low $TREM1^+$ PMN-MDSCs infiltration group, with scoring implemented using the single-sample gene set enrichment analysis (ssGSEA) function. Furthermore, $TREM1$ expression was extracted from each sample. Employing the *surv_cutpoint* function, we calculated the optimal cutoff value and subsequently stratified the samples into high and low expression groups. Subsequently, Kaplan–Meier survival analysis was conducted using the survival R package (v3.2-13) and the survminer R package (v0.5.0). Then, samples were divided into high and low $TREM1$ expression groups based on the median value, with the CIBERSORT algorithm used to evaluate immune cell infiltration characteristics. The marker gene scores of $TREM1^+$ PMN-MDSCs, immunosuppressive function, exhausted T cells, fibroblasts and EMT were also calculated, with subsequent evaluation of their correlations.

## Spatial transcriptomics data analysis

ST data were downloaded from the GEO database to illustrate the spatial distribution of cell types. Following data loading via the Seurat package, normalization was conducted through SCTransform processing. Dimensionality reduction and clustering analyses were analyzed using principal component analysis (PCA). Gene sets encompassing key cellular subtypes (fibroblasts, myeloid cells, $TREM1^+$ PMN-MDSCs, and exhausted T cells) and functional signatures (immunosuppressive function and EMT) were curated for downstream analysis. First, the *AddModuleScore* function was used to calculate the scores for each gene set. Subsequently, the *SpatialPlot* function was employed to visualize the spatial distribution of the aforementioned cell types and functional characteristics. To better illustrate the spatial expression of genes, the resolution of principal components was enhanced using the *spatialEnhance* function from the BayesSpace package (v1.15.3), while gene expression features were further enhanced using the *enhanceFeatures* function[38]. Correlation was interrogated by calculating Pearson correlation coefficients and linear associations were graphically represented through scatter plots. The spatial relationships between cell types were further dissected using the mistyR package (v1.10.0)[39]. First, feature scores of cell-type gene sets were extracted from ST data. Then, the nearest neighbor distances between spots were calculated, and an appropriate radius was selected. Based on this, the analysis was performed through the paraview-spot view. The results were visualized using the *plot_interaction_communities* function to generate spatial spot maps. Furthermore, ST ligand-receptor analysis was performed using stLearn (v0.4.12), with ligand-receptor pairs expressed in fewer than 20 spots being excluded from the analysis. The results were visualized using the *st.pl.lr_result_plot* function[40].

## Cell–cell communication analysis

Intercellular communication across cell types was investigated using the CellChat package (v1.6.1). A CellChat object was initially constructed through the *createCellChat* function, followed by comprehensive analysis based on three major ligand-receptor interaction databases: secreted signaling, ECM-receptor, and cell-cell contact. To infer ligand-receptor-target gene interactions among these cellular subpopulations, we applied the nichenetr package (v2.1.5), with $TREM1^+$ PMN-MDSCs designated as

receivers and endothelial cells, fibroblasts, CD4$^+$ Treg, CD8$^+$ Teff, and NK cell as senders. Genes expressed in at least 10% of cells were included for analysis. Subsequently, the ligands were ranked based on the area under the precision-recall curve (AUPR). From this ranking, the top 28 ligands and their target genes were extracted. Finally, we visualized the interaction network between these ligands and their respective receptors[41].

## Cell culture

The human renal clear cell carcinoma cell line 786-O was purchased from Immocell (Xiamen, China), and the human lung adenocarcinoma cell line H226 was purchased from Procell Company (Wuhan, China). 786-O, H226, and MDSC were cultured in RPMI-1640 medium (Gbico, USA) supplemented with 10% FBS (Gibco, USA) and 1% penicillin/streptomycin (Solarbio, China). T cells were cultured using ImmunoCult™-XF T medium supplemented with *IL-2* and *CD3/CD28* activators (stemcell, China). All cells were cultured at 37 °C in a 5% $CO_2$ incubator.

## T cell and MDSC isolation

T cells were isolated from healthy volunteers' peripheral blood. The procedure is as follows: Collected 10 mL of peripheral blood and diluted it with an equal volume of PBS in a 1:1 ratio. Slowly added 20 mL of human lymphocyte separation medium (Stemcell, China) to the bottom of the centrifuge tube. Centrifuged at 20 °C and $500 \times g$ for 30 min. Gently aspirated lymphocytes from the interface between the separation medium and plasma layer. After washing with PBS, resuspended cells in 1 mL EasySep Buffer (Stemcell, China). Subsequently, T cells were isolated using the $CD3^+$ T cell positive selection kit (Stem Cell Technologies, Inc.). Collected fresh spleens from healthy female C57BL/6J mice ($n = 5$), aged 6–8 weeks, sourced from Guangxi Medical University Animal Center, minced them into a single-cell suspension, and isolated MDSCs using the EasySep™ Mouse MDSC ($CD11b^+Gr1^+$) Isolation Kit (Stemcell, China).

## Co-culture and collection of conditioned medium

786-O and H226 cells were respectively seeded into 6-well plates at a density of $1 \times 10^5$ cells/mL. After cells attachment, MDSCs ($1 \times 10^6$ cells/mL) were added to each well for co-culture. Following 48 h of co-culture, the supernatant was collected to obtain MDSCs treated with tumor cell co-culture. To prepare conditioned medium, both untreated MDSCs and tumor cell co-cultured MDSCs were seeded at $1 \times 10^6$ cells/mL into 6-well plates. Cells were cultured serum-free for 24 h, and the cell culture medium was collected. After centrifugation at 3000 rpm for 5 min to eliminate cell debris, the resulting supernatant was mixed with standard medium at a 1:1 ratio to form conditioned medium.

## CCK8 assay

T cells were seeded into 96-well plates at a density of $1 \times 10^5$ cells/mL, and the culture medium was replaced with the assigned conditioned media. After culturing cells for 24 h, 10 μL of CCK8 reagent (Biyuntian, China) was added to each well. After incubation for 4 h, the absorbance of each group at a wavelength of 450 nm was measured. The detailed results were provided in Supplementary Data 6.

## ELISA assay

T cells were also seeded into 96-well plates at a density of $1 \times 10^5$ cells/mL. After co-culturing cells for 24 h according to the groups, the culture medium was collected and centrifuged at 3000 rpm for 20 min. According to the kit instructions, the supernatant was collected and the levels of TNF-α (ZHUOERYOU, China) and IFN-γ (ZHUOERYOU, China) were measured. Detailed results could be found in Supplementary Data 7.

## Multiplex immunofluorescence staining

mIF staining was performed on paraffin-embedded tissue sections from both tumor and non-tumor samples. Initially, tissue sections were deparaffinized and subjected to antigen retrieval using sodium citrate buffer (pH 6.0) under high temperature conditions. Subsequently, sections were

permeabilized with 0.3% Triton X-100 for 30 min, followed by three 5-min washes with phosphate-buffered saline (PBS). Blocking was done with 10% goat serum for 30 min at room temperature. Then, primary antibodies containing 0.05% Triton X-100 were added and incubated overnight on a rotating platform at 4 °C. After overnight incubation, the sections were washed with PBS, and secondary antibodies containing 0.05% Triton X-100 were added for incubation. Following primary antibody incubation, sections were washed with PBS and incubated with secondary antibodies containing 0.05% Triton X-100. After incubation, nuclei were stained with DAPI (Dojindo Molecular Technologies, 1:500 dilution). Primary antibodies employed in this study included: Anti-*CD11b* antibody (Abcam, ab52478, 1:400 dilution), Rabbit Anti-*LOX1* antibody (Bioss, bs-2044R, 1:400 dilution) and Rabbit Anti-*TREM1* antibody (Bioss, bs-10306R, 1:400 dilution). Immunocomplexes were detected using Alexa Fluor-conjugated secondary antibodies (Life Technologies, 1:400 dilution).

## Statistics and reproducibility
Statistical analyses were performed using GraphPad Prism 9.5 and R software. Pearson correlation analysis was employed to assess associations in bulk RNA-seq and ST data. Inter-group comparisons of continuous variables were conducted using the Wilcoxon rank-sum test. Survival analysis was performed using the Kaplan-Meier method, with inter-group comparisons of survival curves evaluated by the Log-rank test. For experimental data, an unpaired $t$-test was used. A significance level of $p < 0.05$ was adopted for all statistical tests. Significance levels were denoted as follows: $*p < 0.05, **p < 0.01, ***p < 0.001, ****p < 0.0001$.

## Ethical declaration
The acquisition of samples and tissue sections was approved by the Ethics Committee of the First Affiliated Hospital of Guangxi Medical University (approval number: KY-E-097) and the Fifth Affiliated Hospital of Guangxi Medical University (LW2024-DECISION-010). The collection of peripheral blood samples from healthy individuals and spleen samples from C57BL/6J mice was approved by the Ethics Committee of the First Affiliated Hospital of Guangxi Medical University (approval number: 2025-E0787). The patients/participants were properly informed and provided their written informed consent to participate in this study. This study was conducted in accordance with the Declaration of Helsinki. All ethical regulations relevant to human research participants were followed.

## Reporting summary
Further information on research design is available in the Nature Portfolio Reporting Summary linked to this article.

## Data availability
Detailed information on the scRNA-seq and ST datasets, including their sources and accession numbers, can be found in Supplementary Data 1, 3. Among them, raw sequence datasets from our previous studies are stored in the GEO database (accession number: GSE162454) and the Genome Sequence Archive in National Genomics Data Center, China National Center for Bioinformation/Beijing Institute of Genomics, Chinese Academy of Sciences (accession number: HRA007229)[42–45]. Bulk RNA-data and clinical information from TCGA database were incorporated. All other data are available from the corresponding author on reasonable request.

## Code availability
The code script utilized for analysis in this study can be accessed at https://github.com/haijun2595/MDSC.

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

## Acknowledgements

This study was supported by the National Natural Science Foundation of China (82260814, [Y.L.]), the First-class Discipline Innovation-driven Talent Program of Guangxi Medical University ([Y.L.]), "Medical Excellence Award" Funded by the Creative Research Development Grant from the First Affiliated Hospital of Guangxi Medical University ([Y.L.]). We thank Mingjie Chen (Shanghai NewCore Biotechnology Co., Ltd.) for providing data analysis and visualization support.

## Author contributions

Y.L., J.L.H., and S.J.L. designed and supervised the study. Y.J.C., S.H.L., H.N.L., H.J.T., Z.Z., W.Z.W., and P.T.W. collected data. Y.J.C., S.H.L., H.N.L., H.J.T., Z.Z., and K.L. performed data analysis. X.D.Z., H.C.T., M.X.Y., and S.H.L. performed experiments. Y.J.C., S.H.L., and H.J.T. drew the picture. Y.J.C., S.H.L., H.N.L., and Z.Z. wrote the article with the help of X.Y.H. and W.Y.F. All authors contributed to the study design and participated in data interpretation.

## Competing interests

The authors declare no competing interests.
