## [Transparent Peer Review file · Communications Biology]

Pan-cancer analysis reveals TREM1+ PMN-MDSCs as critical regulators of immune suppression and tumor microenvironment remodeling

Corresponding Author: Professor Yun Liu

This manuscript has been previously submitted at another journal. This document only contains information relating to versions considered at Communications Biology.

Version 0:

Reviewer comments:

Reviewer #1

(Remarks to the Author)

Although the research subject is potentially of interest, all data presented in the manuscript are based on in silico analyses without being validated.

- 1) What are the data in the present manuscript that demonstrate the existence of a tumor-associated PMN-MDSC population expressing TREM1?
- 2) Is TREM1 expressed constitutively or is it induced by tumor factors?
- 3) Is TREM1 also expressed in M-MDSCs?
- 4) What is the physiological role of TREM1 in myeloid cells and in particular in PMN-MDSCs?
- 5) Is there a possibility to suppress TREM1 and thus demonstrate its importance in expanding the MDSC population.
- 6) Is TREM1 involved in PMN-MDSC-mediated immunosuppressive role against T cells?

These are some of the basic questions that should be studied and investigated by in vitro and in vivo experimental approach and then try to understand the possible therapeutic application! Therefore, despite the large amount of in silico data, I strongly suggest the authors to validate the main data with in vitro, ex-vivo and in vivo experiments to support their conclusions.

Reviewer #2

(Remarks to the Author)

This study focuses on polymorphonuclear myeloid-derived suppressor cells (PMN-MDSCs), aiming to address key knowledge gaps, including a systematic exploration of their functional heterogeneity across different cancer types and the integration of this heterogeneity with their spatial distribution and interactions with other cellular components in the tumor microenvironment (TME). Overall, the study is well-designed and straightforward, leveraging single-cell RNA sequencing data from 589 samples across 18 cancer types to provide a comprehensive pan-cancer perspective on PMN-MDSCs.

The research identifies a novel PMN-MDSC subpopulation characterized by the upregulation of immunosuppressive genes and marked by TREM1 expression. Notably, the study provides evidence linking the TREM1+ PMN-MDSC subset to clinical outcomes in various malignancies, suggesting its potential as a prognostic biomarker and positioning TREM1 as a possible therapeutic target.

While these findings hold significant promise for cancer research and the development of new treatment strategies, the study has important limitations that diminish its impact. As the authors acknowledge, there is a lack of in-depth mechanistic studies to validate the immunosuppressive functions of the TREM1+ PMN-MDSC subset. This gap, along with other addressable concerns discussed below, currently prevents the manuscript from being suitable for publication.

Specific remarks:

1) The identification of PMN-MDSCs was based on an immunosuppressive gene signature previously defined in tumor-infiltrating monocytic myeloid-derived suppressor cells (M-MDSCs). While the violin plots in Figure 1G (despite potential graphical inaccuracies) show similarly high expression of MDSC signature genes and elevated immunosuppressive scores in both monocyte/macrophage and neutrophil subsets compared to other myeloid cells, this approach raises a critical question: why was an M-MDSC-derived signature used instead of a PMN-MDSC-specific signature? Given that monocytic and granulocytic MDSCs suppress immune responses through both shared and subset-specific mechanisms, it remains unclear whether this gene signature is entirely conserved between the two populations. To address this, a Venn diagram comparing shared and unique genes between monocyte/macrophage and neutrophil subsets could help delineate a PMN-MDSC-specific signature. Such refinement would be particularly valuable for subsequent analyses, which focus exclusively on the PMN-MDSC subset.

2) To assess functional heterogeneity across cancer types, the researchers analyzed an extensive collection of datasets from five different sources, spanning 18 cancer types. However, several key details could enhance clarity:

- Cancer type nomenclature: While Figure 1A and B use icons and acronyms, the full names of each cancer type should be explicitly stated in the text for accessibility.
- Rationale for normal lymph node (NLN) datasets: The inclusion of NLN datasets is unclear unless they serve as controls for metastatic lymph node (Met) samples. If this is the case, the justification should be clarified.
- Improved data visualization: The bar graph in Figure 1B should be modified to display the frequency of each dataset source (Pre, PT, Met, NT, NLN) per tumor type, ensuring transparency in sample composition.

3) Figure 2 B and 2C: As noted for Figure 1G, the statistical comparisons in panels B and C are incorrect. The P0, P2, and P4 subsets exhibit similarly high expression levels of both the “MDSC immunoscore” and “new-MDSC signature”, which are significantly elevated compared to the P1 and P5 subsets. Therefore, the bar graphs and asterisks should explicitly compare:

- P0 vs. P1 and P5
- P2 vs. P1 and P5
- P4 vs. P1 and P5

4) Functional Evolution of PMN-MDSCs During Tumor Progression. While exploring the functional evolution of PMN-MDSCs across tumor stages could yield valuable insights into their mechanistic impact on cancer outcomes, the current experimental design requires refinement. To fully leverage the available datasets—which include samples from precancerous lesions (Pre), primary tumors (PT), and metastases (Met)—I recommend a time-resolved analytical approach that mirrors the natural history of tumor progression:

- NT vs. Pre (early changes during malignant transformation)
- Pre vs. PT (tumor establishment)
- PT vs. Met (progression to metastatic disease)

If the authors retain their current comparison strategy (PT/Met vs. NT), they should at minimum:

1. Include Pre vs. NT to assess baseline alterations in premalignant stages.
 2. Analyze normal lymph nodes (NLN) and metastatic LNs separately, as these represent distinct immunological niches.
- Such stratification would clarify whether PMN-MDSC functional states are acquired progressively or emerge abruptly at specific stages—critical for understanding their clinical relevance.

5) Page 11, lines 323-325. The statement 'These cells are primarily composed of PMN-MDSCs, derived from neutrophils, and M-MDSCs, originating from monocytes' requires correction. Current consensus defines PMN-MDSCs and M-MDSCs as immunosuppressive myeloid cells of granulocytic and monocytic lineages, respectively - not as derivatives of mature neutrophils or monocytes.

While emerging evidence suggests that PMN-MDSCs may potentially arise from peripheral neutrophil reprogramming in tumor microenvironments, the predominant view supported by most experimental data indicates that both PMN-MDSCs and M-MDSCs represent developmentally arrested immature myeloid cells. The exact nature of PMN-MDSCs - whether they constitute a distinct maturation-arrested population or alternatively represent a pathological functional state that can be acquired by more mature neutrophils - remains an active area of investigation.

Version 1:

Reviewer comments:

Reviewer #1

(Remarks to the Author)

Most of the raised questions have been properly addressed. Although the authors were unable to directly verify the role of TREM1 in the expansion of MDSC population by inhibiting TREM1, a study conducted by Ashwin Ajith et al. provides strong evidence that silencing TREM1 can restrict the expansion of MDSCs. The new version of the manuscript is sufficiently improved.

Reviewer #2

(Remarks to the Author)

Dear Authors,

Thank you for your thorough revisions and detailed responses to my comments. You have done an excellent job addressing all of my concerns. The manuscript is much improved, and I recommend it for acceptance.

Dear reviewers:

Thanks again for your kind work and valuable comments. On half of my co-authors, we would like to express our great appreciation to you. Also, we have studied your comments carefully and have made correction. Revised portion are marked in red in the manuscript. The main corrections in the paper and the responds to reviewers' comments are as flowing:

Responds to the comments of reviewer1:

1. Comment: *What are the data in the present manuscript that demonstrate the existence of a tumor-associated PMN-MDSC population expressing TREM1?*

Response: Thank you for your valuable comment. We have supplemented the analysis of *TREM1* expression in PMN-MDSCs across different groups. The results indicate that the *TREM1* gene was highly expressed in the tumor-associated PMN-MDSC population. The corresponding data have been added to **Fig. 3C**.

2. Comment: *Is TREM1 expressed constitutively or is it induced by tumor factors?*

Response: Thanks for your suggestion. In our additional analysis, we observed that *TREM1* is highly expressed in tumor-associated PMN-MDSCs. Furthermore, we utilized an online analysis platform (<http://www.bioinformatics.com.cn/>) to examine *TREM1* expression across multiple cancer types, and the results indicate that this gene is significantly upregulated in various tumor tissues, as shown in the figure below. Additionally, our multiplex immunofluorescence experiments also demonstrated markedly higher *TREM1* expression in tumor tissues compared to non-tumor tissues, as presented in **Fig. 4J**. Therefore, we propose that *TREM1* is not only constitutively expressed but can also be significantly induced by tumor factors, leading to a higher expression level.

The expression of the *TREM1* gene in multiple cancers (cited from <http://www.bioinformatics.com.cn/>).

3. Comment: *Is TREM1 also expressed in M-MDSCs?*

Response: Thank you for raising this important question. Following the associate reviewers' comments, we performed new analysis. Based on M-MDSC-specific gene sets derived from two literature sources (PMID:39818911, 32086381), we analyzed monocyte/macrophage subpopulations to identify M-MDSC cells within them. Results indicate that the M5_FABP4 subpopulation was identified as a potential M-MDSC. In our study, the *TREM1* gene was highly expressed in the M-MDSC cell subpopulation, as shown in the figure below. Furthermore, our literature review revealed that *TREM1* expression has been reported in monocytic MDSCs (Mo-MDSCs) in tumor-bearing mice (PMID: 19740375, 35223446).

Identification of M-MDSCs. **A.** UMAP of monocyte/macrophage subsets. **B.** Violin plot showing scores for the M-MDSC-specific gene set in monocyte/macrophage subpopulations. **C.** Violin plot showing *TREM1* gene expression.

4. Comment: *What is the physiological role of TREM1 in myeloid cells and in particular in PMN-MDSCs?*

Response: Thank you for this insightful question. We examined the expression pattern of *TREM1* across myeloid cells and found it to be highly expressed in monocytes/macrophages and neutrophils. We specifically focused on the functional

enrichment of *TREMI* within these subsets, with the results presented in the figure below. Functional enrichment analysis revealed that the *TREMI* gene is primarily involved in key biological processes including the humoral immune response, acute inflammatory response, and scaffold protein binding. Furthermore, we stratified PMN-MDSCs into *TREMI*-high and *TREMI*-low groups based on the 70th percentile of *TREMI* expression. Differential gene expression analysis between these groups showed that upregulated genes in the *TREMI*-high group were significantly enriched in pathways such as reactive oxygen species response, cell chemotaxis, and regulation of angiogenesis. In contrast, differentially expressed genes in the *TREMI*-low group were predominantly enriched in pathways related to ribosome biogenesis, assembly, and processing. The corresponding results have been added to **Supplementary Fig. 2C**.

***TREMI* expression and functional enrichment in myeloid cells.** **A.** Expression levels of the *TREMI* gene across myeloid cell subsets. **B.** Bar plot illustrating functional pathways significantly associated with *TREMI* activation.

5. Comment: *Is there a possibility to suppress TREMI and thus demonstrate its importance in expanding the MDSC population.*

Response: We sincerely appreciate the profound and valuable suggestion put forward by the reviewer. Due to the limitations of the current experimental conditions, we were unable to conduct experiments in this study to directly verify the role of *TREMI* in the expansion of MDSC population by inhibiting *TREMI*. However, we fully agree that this experiment is crucial for confirming the functional association between *TREMI* and MDSC expansion. To address the reviewer's concern, we have conducted an in-depth literature search. A study conducted by Ashwin Ajith et al. provides strong evidence for this. By using *Trem1*-deficient mice and combining single-cell RNA

sequencing with functional assays, this study confirmed that silencing *TREMI* can restrict the expansion of MDSCs and weaken their immunosuppressive capacity (PMID: 37651197). This result is highly consistent with the hypothesis proposed by the reviewer, indicating that silencing *TREMI* can limit the expansion of MDSC population. We would like to thank the reviewer again for the insightful suggestion. It not only helps us further improve the logic of our research but also provides an important direction for subsequent exploration of the specific mechanism of *TREMI* in MDSC regulation, thereby broadening our research perspective.

6. Comment: *Is TREMI involved in PMN-MDSC-mediated immunosuppressive role against T cells?*

Response : We would like to thank the reviewer for this important comment. In this study, functional co-culture experiments were performed using the total MDSC population isolated from mouse spleens. We acknowledge that this population is a mixture; however, it is predominantly composed of PMN-MDSCs. Based on this cellular composition, we infer that the observed immunosuppressive effects of the total MDSCs are primarily driven by the PMN-MDSC subset. Specifically, to evaluate the impact of MDSCs on T cell function, we conducted co-culture experiments by incubating T cells with MDSCs isolated from the spleens of C57BL/6J mice. The results showed that, compared to the T cell group, co-culture with MDSCs significantly inhibited T cell activity and the secretion levels of IFN- γ and TNF- α . Furthermore, it was found that T cell function was more significantly inhibited after T cells were co-cultured with *TREMI*⁺ MDSCs and tumor cells. This confirms that MDSCs exert inhibitory effects via *TREMI*. The corresponding results have been added to **Fig. 4K**. We are grateful for the reviewer’s suggestion, which has helped us refine our work to a higher standard.

K. Compared with the T cell group, the function of T cells was inhibited after co-culture with MDSCs. When MDSCs co-cultured with tumor cells were added to T cells, the inhibition of T cell function became more significant.

Responds to the comments of reviewer2:

1. **Comment:** *The identification of PMN-MDSCs was based on an immunosuppressive gene signature previously defined in tumor-infiltrating monocytic myeloid-derived suppressor cells (M-MDSCs). While the violin plots in Figure 1G (despite potential graphical inaccuracies) show similarly high expression of MDSC signature genes and elevated immunosuppressive scores in both monocyte/macrophage and neutrophil subsets compared to other myeloid cells, this approach raises a critical question: why was an M-MDSC-derived signature used instead of a PMN-MDSC-specific signature? Given that monocytic and granulocytic MDSCs suppress immune responses through both shared and subset-specific mechanisms, it remains unclear whether this gene signature is entirely conserved between the two populations. To address this, a Venn diagram comparing shared and unique genes between monocyte/macrophage and neutrophil subsets could help delineate a PMN-MDSC-specific signature. Such refinement would be particularly valuable for subsequent analyses, which focus exclusively on the PMN-MDSC subset.*

· **Issue 1:** *why was an M-MDSC-derived signature used instead of a PMN-MDSC-specific signature?*

Response: We appreciate the thoughtful review and constructive feedback provided by the reviewer. We fully agree that using population-specific signatures to define PMN-MDSCs is crucial. In the original manuscript, the gene signature used in **Fig. 1G** to define MDSCs within myeloid subsets was derived from studies by Tsutsumi et al. (PMID: 38217732) and Alshetaiwi et al. (PMID: 32086381), and it was subsequently applied to identify PMN-MDSCs. We have now updated **Fig. 1G** by reapplying the intersecting genes of PMN-MDSC and M-MDSC reported in the study by Tsutsumi et al. The revised violin plot more accurately reflects the presence of MDSCs within the myeloid compartment.

To precisely identify PMN-MDSCs within the neutrophil cluster, we have now used a combined approach incorporating the canonical PMN-MDSC gene set reported by Tsutsumi et al. together with an immunosuppressive gene set. This strategy ensures a more specific definition of the PMN-MDSC subpopulation. The results are now presented in the updated **Fig. 2B** and **Fig. 2C**. We believe these modifications have significantly enhanced the accuracy of our analysis, and we thank the reviewer for prompting this important refinement.

• **Issue 2:** *To address this, a Venn diagram comparing shared and unique genes between monocyte/macrophage and neutrophil subsets could help delineate a PMN-MDSC-specific signature. Such refinement would be particularly valuable for subsequent analyses, which focus exclusively on the PMN-MDSC subset.*

Response: Similarly, following the reviewer's suggestion, we compared the upregulated genes between monocyte/macrophage and neutrophil subpopulations using a Venn diagram and found that only the common gene *OLRI* existed (Venn diagram below). It may be difficult to obtain a sufficient number of specific genes to define a reliable signature by comparing the differences between monocyte/macrophage and neutrophil subpopulations.

2.Comment: *To assess functional heterogeneity across cancer types, the researchers analyzed an extensive collection of datasets from five different sources, spanning 18 cancer types. However, several key details could enhance clarity:*

- *Cancer type nomenclature: While Figure 1A and B use icons and acronyms, the full names of each cancer type should be explicitly stated in the text for accessibility.*
- *Rationale for normal lymph node (NLN) datasets: The inclusion of NLN datasets is unclear unless they serve as controls for metastatic lymph node (Met) samples. If this is the case, the justification should be clarified.*

• *Improved data visualization: The bar graph in Figure 1B should be modified to display the frequency of each dataset source (Pre, PT, Met, NT, NLN) per tumor type, ensuring transparency in sample composition.*

• **Issue 1:** *Cancer type nomenclature: While Figure 1A and B use icons and acronyms, the full names of each cancer type should be explicitly stated in the text for accessibility.*

Response: We thank the reviewer for raising these important points. We fully agree that providing the full names of all cancer types in the main text would improve the manuscript's accessibility. Following your recommendation, we have now included a complete list of all cancer types studied, along with their corresponding acronyms, in the "Abbreviations" section of the main text. Additionally, we have provided a detailed table containing the complete nomenclature of cancer types in the Supplementary Materials for further reference.

• **Issue 2:** *Rationale for normal lymph node (NLN) datasets: The inclusion of NLN datasets is unclear unless they serve as controls for metastatic lymph node (Met) samples. If this is the case, the justification should be clarified.*

Response: We sincerely appreciate the reviewers for highlighting this issue and apologize for any confusion caused by the insufficient justification for including the normal lymph node (NLN) dataset in our original manuscript. Following the recommendation, we have excluded a small number of normal lymph node (NLN) sample and precancerous samples from our analysis. We subsequently re-analyzed the remaining 576 samples (including normal tissues, primary tumors and metastatic lesions). This adjustment did not alter the study's core conclusions. All relevant results, figures in the manuscript have been comprehensively updated based on the new analysis. Thank you for your valuable comments. This revision has significantly improved the preciseness of our research and the reliability of our conclusions. Once again, we would like to express our sincere thanks for your valuable suggestions.

• **Issue 3:** *Improved data visualization: The bar graph in Figure 1B should be modified to display the frequency of each dataset source (Pre, PT, Met, NT, NLN) per tumor type, ensuring transparency in sample composition.*

Response: We sincerely thank the reviewer for this valuable suggestion. Following the recommendation, we have revised **Fig. 1B** by replacing the original aggregated presentation with a grouped bar chart segmented by tumor type. The new figure clearly present the frequency distribution of different sample sources (PT, Met, NT) within each tumor type, ensuring complete transparency in sample composition. Thank you again for helping us enhance the clarity of our data presentation.

3.Comment: *Figure 2 B and 2C: As noted for Figure 1G, the statistical comparisons*

in panels B and C are incorrect. The P0, P2, and P4 subsets exhibit similarly high expression levels of both the “MDSC immunoscore” and “new-MDSC signature”, which are significantly elevated compared to the P1 and P5 subsets. Therefore, the bar graphs and asterisks should explicitly compare:

P0 vs. P1 and P5

P2 vs. P1 and P5

P4 vs. P1 and P5

Response: We sincerely thank the reviewer for the reminder. Following your suggestion, we have re-analyzed the statistical comparisons in **Fig.2B** and **Fig. 2C**. The statistical annotations in the figures now explicitly show the comparison results between the four highest-scoring subpopulations (P0_CXCL3, P2_TREMI, P3_IL1B, and P4_IFIT2) and other subpopulations. We have updated all relevant statistical annotations and ensured that the descriptions in the main text are fully consistent with the updated figure results. We appreciate your comment that has helped us improve the accuracy of our study.

4.Comment: *Functional Evolution of PMN-MDSCs During Tumor Progression. While exploring the functional evolution of PMN-MDSCs across tumor stages could yield valuable insights into their mechanistic impact on cancer outcomes, the current experimental design requires refinement. To fully leverage the available datasets—which include samples from precancerous lesions (Pre), primary tumors (PT), and metastases (Met)—I recommend a time-resolved analytical approach that mirrors the natural history of tumor progression:*

- *NT vs. Pre (early changes during malignant transformation)*
- *Pre vs. PT (tumor establishment)*
- *PT vs. Met (progression to metastatic disease)*

If the authors retain their current comparison strategy (PT/Met vs. NT), they should at minimum:

- 1. Include Pre vs. NT to assess baseline alterations in premalignant stages.*
- 2. Analyze normal lymph nodes (NLN) and metastatic LNs separately, as these represent distinct immunological niches.*

Such stratification would clarify whether PMN-MDSC functional states are acquired progressively or emerge abruptly at specific stages—critical for understanding their clinical relevance.

· **Issue 1:** *To fully leverage the available datasets—which include samples from precancerous lesions (Pre), primary tumors (PT), and metastases (Met)—I recommend a time-resolved analytical approach that mirrors the natural history of tumor progression:*

- *NT vs. Pre (early changes during malignant transformation)*

- *Pre vs. PT (tumor establishment)*

- *PT vs. Met (progression to metastatic disease)*

Response: We sincerely appreciate the reviewer's important suggestion, and the issue you raise is of great significance to us. We fully agree on the value of time-series analysis for deciphering the functional evolution of PMN-MDSCs during tumor progression. In our analysis, to maintain cohort homogeneity and analytical rigor, we excluded a small number of normal lymph node samples and precancerous lesion samples from the dataset before re-analysis. To further address your suggestion, we focused our analysis on comparing primary tumors (PT) with metastases (Met) using the existing sample set comprising primary tumors, metastases and normal tissue (NT), thereby simulating the progression from primary to metastatic sites. These results are integrated into **Figure 3**. We believe this optimized analysis strategy, while constrained by existing sample types, effectively reveals key evolutionary patterns in the functional state of PMN-MDSCs. We sincerely thank the reviewer for these valuable suggestions, which have significantly enhanced the depth and rigor of our research.

· **Issue 2:** *If the authors retain their current comparison strategy (PT/Met vs. NT), they should at minimum:*

1. *Include Pre vs. NT to assess baseline alterations in premalignant stages.*

Response: We fully agree that evaluating differences between Pre and NT group is crucial for establishing baseline changes. However, in the current cohort of this study, the number of available precancerous lesion samples was extremely limited after rigorous quality control and screening. We are concerned that such a small sample size may be insufficient to yield statistically reliable and biologically generalizable conclusions, potentially introducing bias instead. Therefore, in the interest of scientific rigor, we no longer insist on the original PT/Met vs. NT comparison strategy. Instead, we fully adopt the reviewers' suggestion and shift the analytical focus to the PT vs. Met comparison, which simulates the tumor progression process.

· **Issue 3:** *Analyze normal lymph nodes (NLN) and metastatic LNs separately, as these represent distinct immunological niches.*

Response: We agree with the reviewer's point. As detailed in our response to Issue 2, we have moved away from the broad PT/Met vs. NT comparison. Consequently, our primary analysis now focuses on a series of direct comparisons that model tumor progression: PT vs. Met. We extend our gratitude once again to you for your thoughtful guidance.

5.Comment: Page 11, lines 323-325. The statement 'These cells are primarily composed of PMN-MDSCs, derived from neutrophils, and M-MDSCs, originating from monocytes' requires correction. Current consensus defines PMN-MDSCs and M-MDSCs as immunosuppressive myeloid cells of granulocytic and monocytic lineages, respectively - not as derivatives of mature neutrophils or monocytes. While emerging evidence suggests that PMN-MDSCs may potentially arise from peripheral neutrophil reprogramming in tumor microenvironments, the predominant view supported by most experimental data indicates that both PMN-MDSCs and M-MDSCs represent developmentally arrested immature myeloid cells. The exact nature of PMN-MDSCs - whether they constitute a distinct maturation-arrested population or alternatively represent a pathological functional state that can be acquired by more mature neutrophils - remains an active area of investigation.

Response : We appreciate the opportunity to provide further clarification on this important point. We acknowledge that the original wording was imprecise. Following your suggestion, we have now revised the relevant description to align with the current consensus (PMID: 38523155). The original text has been modified to 'These cells are primarily composed of PMN-MDSCs, derived from the granulocytic lineages, and M-MDSCs, originating from the monocytic lineages'. Relevant modifications were updated in Discussion section (page 12, lines 354-356). We acknowledge your comments and constructive suggestions very much, which are valuable in improving the quality of our manuscript.